# Fast-Growing Alveolar Echinococcosis Following Lung Transplantation

**DOI:** 10.3390/pathogens9090756

**Published:** 2020-09-16

**Authors:** Clarisse Dupont, Fréderic Grenouillet, Jean-Yves Mabrut, Frédérique Gay, Florence Persat, Martine Wallon, Jean-François Mornex, François Philit, Damien Dupont

**Affiliations:** 1Service de Pneumologie, Hôpital Louis Pradel, Hospices Civils de Lyon, 69500 Bron, France; clarisse.dupont@chu-lyon.fr (C.D.); jean-francois.mornex@chu-lyon.fr (J.-F.M.); 2Unité de Sérologies Parasitaires et Fongiques, Laboratoire de Biologie Médicale, Centre Hospitalier Régional Universitaire, 25000 Besançon, France; fgrenouillet@chu-besancon.fr; 3UMR6249 CNRS-UBFC Chrono-Environnement, Université de Bourgogne-Franche-Comté, 25000 Besançon, France; 4Service de Chirurgie Digestive et de Transplantation Hépatique, Hôpital de la Croix-Rousse, Hospices Civils de Lyon, 69317 Lyon, France; jean-yves.mabrut@chu-lyon.fr; 5Inserm, CRCL UMR1052, Université de Lyon, Université Lyon 1, 69373 Lyon, France; 6Service de Radiologie, Hôpital Edouard Herriot, Hospices Civils de Lyon, 69003 Lyon, France; frederique.gay@chu-lyon.fr; 7Institut des Agents Infectieux, Parasitologie Mycologie, Hôpital de la Croix-Rousse, Hospices Civils de Lyon, 69317 Lyon, France; florence.persat@chu-lyon.fr (F.P.); martine.wallon@chu-lyon.fr (M.W.); 8Physiologie Intégrée du Système D’éveil, Centre de Recherche en Neurosciences de Lyon, INSERM U1028-CNRS UMR 5292, Centre Hospitalier Le Vinatier, Université de Lyon, Université Lyon 1, 69675 Bron, France; 9INRAE, IVPC, UMR754, Université de Lyon, Université Lyon 1, 69007 Lyon, France; 10Service de Pneumologie, Hôpital de la Croix Rousse, Hospices Civils de Lyon, 69317 Lyon, France; francois.philit@chu-lyon.fr

**Keywords:** *Echinococcus multilocularis*, parasite, liver, lung transplantation, immunosuppression

## Abstract

Alveolar echinococcosis is a rare but life-threatening infection caused by the parasite *Echinococcus multilocularis*. Its natural history is characterized by a slow parasitic growth over several years. Increased incidence and shorter development delay have been reported in immune-compromised patients. We report the reactivation of aborted lesions within 12 months of lung transplantation leading to a fast-growing aggressive hepatic lesion. Timely identification of alveolar echninococcosis allowed prompt albendazole treatment and radical surgery leading to a favorable outcome 42 months after transplantation. However, close clinical, serological and radiological monitoring is required to rule out relapses in the long term. The pre-existence of aborted self-limited lesions of alveolar echinococcosis and the possibility for their atypical rapid growth in patients undergoing profound immunosuppression should be known by healthcare providers, even if working in non-endemic areas.

## 1. Introduction

Alveolar echinococcosis (AE) is a rare but potentially life-threatening infection due to the accidental ingestion of the egg of the parasite *Echinococcus multilocularis*. It is encountered only in the northern hemisphere [1]. The natural history of infection in humans potentially follows three different scenarios depending on the immune status of the patient that has ingested *E. multilocularis* eggs: (i) seroconversion proving infection, either without associated liver damage (failure of parasite development) or presenting with a calcified lesion (aborted lesion) (ii) conventional AE in patients for whom the immune system partially controls the development of the parasite’s larva (metacestode), which will present clinical signs five to 15 years after infestation and (iii) uncontrolled hyperproliferation of the metacestode in immunodeficient patients [2]. An increased incidence of AE in immunosuppressed patients has been reported over the past decade, first in France then in Switzerland, where immunocompromised patients accounted, respectively, for 18% and 31% of newly-diagnosed AE cases [3,4,5]. Physicians are now confronted with this epidemiological shift and must now consider the risk of developing EA during an immunosuppressive regimen/pathology, either by de novo contamination or by reactivation of a preexisting lesion that has gone undetected [3,5]. Thus, small calcified lesions which were considered as aborted might represent a potential threat in this context, as they could play the role of a Trojan horse, leading to rapid metacestode proliferation.

## 2. Case

We report an exceptionally fast growing and aggressive AE in a 41-year old Caucasian female veterinarian who underwent a right lung transplantation (LT) for pulmonary fibrosis. In addition to tacrolimus, mycophenolate and prednisolone started immediately post-transplantation, basiliximab—an interleukin 2 receptor antagonist—was required two months post-transplantation for persistent acute cellular rejection treated with intravenous methylprednisolone since one month post-transplantation. Ten months after transplantation, irregular hepatomegaly was detected associated with persisting pain in the right hypochondrium. An abdominal scan performed twelve months after transplantation revealed a 44 mm lesion in the hepatic segment IV with satellite nodules and necrotic adenomegaly suggesting cholangiocarcinoma (Figure 1).

Fortunately, a radiologist, experienced with alveolar echinococcosis, reoriented the diagnosis, based on the heterogeneous aspect of the lesion on the magnetic resonance imaging (MRI) and peripheral contrast and inflammatory perilesional variation and particularly multiple peripheral infracentrimetrical vesicles seen on axial T2-weighted MRI (Figure 1). A Positron Emission Tomography-Computed Tomography (PET/CT) scan confirmed the existence of a 45 mm by 35 mm hepatic lesion with a central hypodense region and a strong peripheral hypermetabolic rim (Standardized uptake values (SUV) max 5.3), persisting at 3 h (SUV max 7.7) and the absence of other associated lesions. Microscopic examination of a liver biopsy fragment showed necrosis and the presence of typical periodic acid-Schiff (PAS) positive thin lamellar layer. Blood tests with Em2+ and Em18 enzyme-linked immunosorbent assay (ELISA) (Bordier, Affinity Products, Crissier, Switzerland) gave positive results. Immunoblotting (Echinococcosis Western blot kit, LDBIO Diagnostics, Lyon, France) was also positive and supported the diagnosis of AE, showing the presence of 7-16-18-26/28 bands, characteristic pattern of *E. multilocularis* infection.

Treatment by albendazole (400 mg bid orally) was immediately initiated. Hepatectomy and right adrenalectomy were performed 15 months post-transplantation with free resection margins. Pathological examination confirmed the diagnosis of AE [6], showing an unequivocal pseudotumoral lesion with numerous small vesicles and ill-defined borders, multiple areas of necrosis and granuloma surrounding slender parasitic PAS-positive layers [7]. At the last examination 42 months after transplantation, the patient was asymptomatic with unremarkable clinical examination and imaging (including 3 h 18-FDG post-injection PET/CT) and undetectable (Em2+) or weak and declining (Em18) serology levels.

Oral albendazole was uninterrupted since diagnosis. Regular monitoring of albendazole sulfoxide plasma levels allowed to reduce the daily posology to 200 mg bid in order to maintain concentrations within the target range (values ranging from 2.30 to 3.30 µmol/L). It was well-tolerated and not interacting with the immunosuppressive regimen.

The fast development of the parasitic lesion in our patient was confirmed by the retrospective *Echinococcus* serology of a serum sampled at the time of transplantation. It was found to be negative for first line tests (ELISA, Indirect Hemagglutination Assay) but immunoblotting showed the presence of a weak seven-band, compatible with aborted, self-limited alveolar echinococcosis. In support to this hypothesis, reexamination of abdominal computed tomography scans two months after transplantation revealed the presence of a small low density nodule within the liver, associated with a small central calcification (Figure 2).

The patient lives in a rural part of the French Massif Central region (France), a historic endemic zone for alveolar echinococcosis [8,9]. Direct interrogation at the time of diagnosis found exposure to several risk factors for AE prior to her transplantation and thereafter [10]: ownership of two dogs and four cats that were dewormed regularly but were free to roam outdoors and hunt unattended including in her unfenced garden visited by foxes; living in a house close to fields; vocational activities in the forest and growing leaf and root vegetables. She worked as a veterinarian for pets prior to transplantation but did not resume work before seven months post transplantation. She reported washing produces from her garden and good hand hygiene after handling and playing with dogs and cats. Alveolar echinococcosis has become less frequent in the Massif Central over the last two decades; therefore, this particular parasitic etiology was not at the forefront during differential diagnosis.

## 3. Discussion

Human alveolar echinococcosis is characterized by the slow growth of lesions under normal circumstances, and clinical signs in immunocompetent hosts are usually not detected before five to 15 years post infection [11]. Increased susceptibility and shorter development times, averaging four years, have been reported in patients presenting with a variety of immunosuppressive conditions including solid cancer, chronic inflammatory disease, malignant hematological disorder, solid organ transplantation and AIDS [3]. Cases of fast growing lesions (less than three to four years) were published in kidney or heart transplant recipients [12,13], but there have been none so far in the context of lung transplantation, and of pre-existing presumptive aborted lesions. Aborted lesion is defined by the presence of a non-viable parasitic structure as evidenced by imaging, revealing a completely calcified lesion [2,14]. However, obtaining parasitological evidence of a non-viable and “dead” lesion is almost impossible for ethical reasons. Spontaneous death of the parasite in the aborted lesion is therefore presumed, but impossible to actually prove [10]. The profound immunosuppression required after lung transplantation and the drugs administered one month post transplantation to avoid acute cellular rejection, probably favored in our patient the rapid and aggressive reactivation of the aborted lesions, retrospectively found to be present two months post transplantation. The natural history of AE is the consequence of a balance between the growth of the metacestode and the host’s response, which depends on the genetic background of the host as well as on acquired disturbances of the Th1-related immune response. Th1-oriented immune response induces protective immunity, leading eventually to self-limited lesion. The laminated layer plays a major role in the induction of tolerance, driven by the Th2 response and anti-inflammatory cytokines, especially IL-10 and TGF-β [2,15]. Our patient received several immunosuppressive treatments that significantly impact on the Th1/Th2 balance and thus the host–parasite interaction, mainly tacrolimus—a calcineurin inhibitor known to inhibit the T-cell response and decrease the Th1/Th2 ratio—and basiliximab. A small lesion with central calcification, as observed in our patient, fits with the description of CT type IV lesion using *Echinococcus multilocularis* classification for computed tomography (EMUC-CT) [16]. Some authors hypothesize that this stage corresponds to the initial phase of the infection, with a strong immune response around the lesion to contain the infection, leading to the persistence of a self-limited dormant infection. Subsequently, this stage can evolve either towards an active infection (EMUC stage I to III) or involute, with necrosis and/or increased calcification [16,17]. They can have different aspects but usually include one or several small size calcifications. Active AE imaging patterns vary from small cystoid lesions to primarily circumscribed tumor-like or diffuse infiltrating lesions with various degrees of calcifications observed using a CT-scan. The pathognomonic honeycomb-like AE microcysts were shown using T2 weighted MRI [10,16]. The FDG–PET/CT often shows increased uptake of FDG surrounding AE lesions, higher than in other areas (as in our patient), and has thus become the preferred reference tool for assessing their metabolic activity [1,10]. These characteristic images, as well as the telltale signs of an active infection, and the possibility for both to exhibit atypical rapid growth in solid organ transplant should be known by healthcare providers [18], even if working in non-endemic areas. Interrogation is important to elicit recent or past long term residence in areas of parasite transmission and work- or life style-related risk factors. The worldwide extension of areas of active transmission and the possible resurgence in historic foci, should lead to an increased awareness, and to listing AE among the etiologies for liver lesions in national guidelines regarding transplantation and other contexts involving immunosuppression, wherever epidemiologically relevant.

Not dismissing the existence of an alveolar echinococcosis is all the more important because it is a differential diagnosis of cholangiocarcinoma, or metastasis of cancer with the primary site being the liver. Chemotherapy that would have been prescribed for cancer etiologies, would facilitate parasitic invasion or reactivation.

Searching for self-limited alveolar echinococcosis, or confirming active lesions in immunocompromised patients, requires both radiological and serological expertise. Serological work-up should systematically include immunoblotting in such patients to maximize sensitivity, since first line tests may be falsely negative in such context [1,3,19]. In immunocompromised patients, first-line techniques (ELISA) have significantly lowered sensitivity and Em-immunoblotting appears as the most sensitive test to diagnose AE [19]. Similarly, a recent review on laboratory-based approaches for AE diagnosis proposed the implementation of immunoblotting as a second line test in a patient with a compatible lesion shown by imaging techniques, but with negative serology using first-line ELISA [20]. This review also highlights the interest of histological examination and/or molecular techniques on liver biopsy, particularly in the same group of patients (negative serology but lesion shown with MRI/CT or ultrasonography). Even if liver biopsy is not recommended as a first line approach, fine-needle biopsy is a minimally invasive and effective diagnostic to confirm hepatic AE [6,21]. Moreover, liver biopsy is not associated with major complications in AE patients. Histological examination of tissue, showing characteristic PAS-positive metacestode layer allows for the diagnosis of AE, even if differential diagnosis with cystic echinococcosis is sometimes difficult. Thus, it should require the implementation of an immunohistochemistry assay using mAb Em2G11 staining [7,22].

AE diagnosed in the context of immunosuppression has a poor prognosis overall [3]. In the case of our patient, both timely parasitostatic treatment and rapid radical surgery were decisive.

Close clinical, serological and radiological post-intervention follow-up will remain essential; even after the end of treatment planned for two years post-alveolar echinococcosis surgery at the earliest [6], or until the serology becomes negative. It is even the more important in the case of our patient as she opted to continue to work as a veterinarian, providing care for mainly house pets, in an endemic alveolar echinococcosis area.

Clinicians attending to immunosuppressed patients living in endemic areas should not only be aware of the need to search for preexisting lesions, but also help prevent new cases of infections. Information should more specifically target dog owners [10] and could be relayed by veterinarians and pet groomers, with the support of educational flyers or posters addressing the need for regular deworming, for preventing roaming and for good hand hygiene after any contact.

## Figures and Tables

**Figure 1 pathogens-09-00756-f001:**
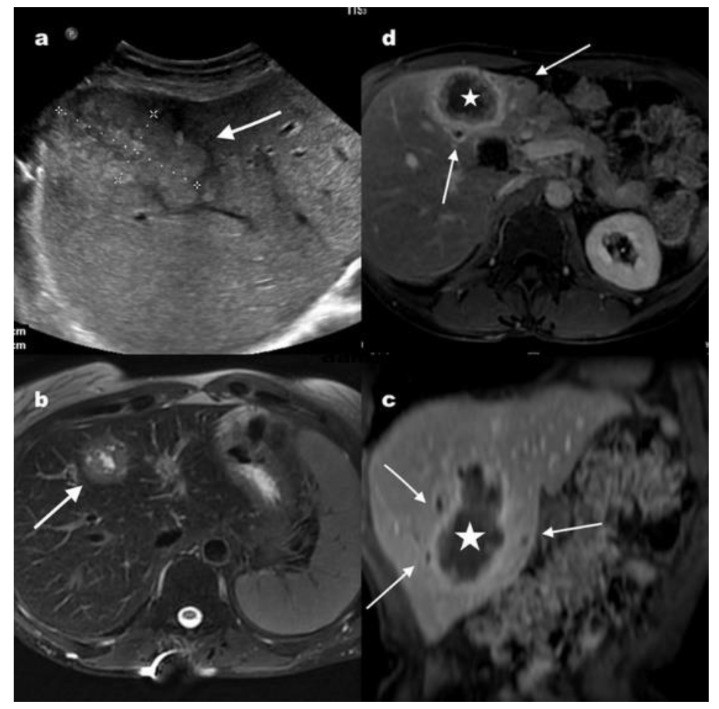
Radiological findings at diagnosis. (**a**) Abdominal gray-scale ultrasound showing a heterogeneous mass lesion in the right lobe of the liver (white arrows). The lesion is generally hyperechoic with a central linear hypoechoic suggesting liquid component. There are also small hyperechoic nodules surrounding this main lesion. (**b**) Axial T2 weighted image after fat suppression demonstrates a tumor like hepatic mass (arrow), heterogeneous with a thick irregular border and central liquid component. (**c**) Axial and coronal (**d**) T1-weighted images with fat suppression, obtained after injection of gadolinium-based contrast agent showing a large heterogeneous tumor like mass in the liver (star), with intense peripheral enhancement and inflammation of adjacent liver. There are also satellite small hypointense vesicles with moderate peripheral contrast enhancement (arrows).

**Figure 2 pathogens-09-00756-f002:**
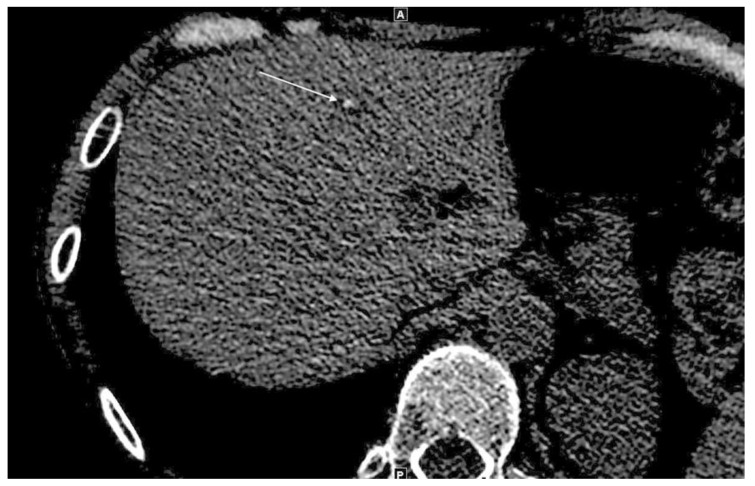
Calcified lesion in the liver two months after transplantation: Axial unenhanced CT showing a small low-density nodule within the liver and the development of a small central calcification within the lesion (arrow).

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
