# Peer review of "Fast-Growing Alveolar Echinococcosis Following Lung Transplantation"

_pathogens, 2020, doi:10.3390/pathogens9090756_

Round 1

Reviewer 1 Report

This is an exciting case report, not only as it demonstrates the fact that immunosuppression leads to an accelerated growth of the parasite Echinococcus multilocularis, but more importantly, that parasitic lesions - identified as "aborted" or "non-viable"- also labelled as "died-out" in the literature - can be re-activated under certain circumstances. Immunosusuppression and anti-GvHD treatment in this case led to a very fast growth of a parasitic lesion and demonstrates that "aborted lesions" can be still viable in humans. This finding may lead to reconsideration of immuological events after infection.

The paper is fluently written and expands on many aspects of the disease. However, some of the statements are more general and are not well referenced. Authors may decide to skip those parts and remain to the main results, or to expand with full references.

Specific Comments:
L 52: Explain the antibody basiliximab
L 74: Histopathologically, AE was confirmed! This must be expressed. Liver biopsy is not recommended if the AE diagnosis is around! Pls. discuss later.
L 77: IHA is an unspecific test using E. granulosus antigen, often used as a screening test. It should not be stressed that the test was negative. This tells nothing. Specific tests are positive, and this is interesting as - often - unter immunosuppression those test turn out to be negative!! See Geyer et al. Discuss this fact.
L 79: Diagnosis was confirmed by histopathology (see Brunetti et al, or Kern et al (not cited). Specific staining could have made the diagnosis more obvious (Barth et al.). Perhaps some additional words in the discussion? Could be an interesting issue.
L 84: Possibly, a PET/CT scan was not performed at the time of diagnosis? Should be mentioned why it was not considered (see Brunetti et al, Kern et al, Wen et al). Too costly, different clinical environment?
L 87: Drug levels are available. I am suggesting to add a table with the medication and level, as this gives much help to clinicians if confronted with similar cases. could also be discussed. Increased incidences of immunosuppression and AE is anticipated in the article of Chauchet et al.!
L 92: I am not convinced that such a weak band in IB would be suggestive of an abortive AE! Sera were checked in retrospect and with knowledge of the outcome! ELISA was negative. This should be the information of importance for readers and needs a change of the argumentation in Discussion.
L 100: I miss sentences regarding the activities of the veterinarian patient before and after tx. Did she has contact to her dogs while under immunosuppression? New- or re-infection issue needs to be discussed!! Had dogs/cats been treated with PZQ regularily, i.e. 6 weeks or around? Did she wash the vegetables from their garden? Statements in the paragraph are rather general and not adapted to the knowledge from literature (risk factors for AE, Kern et al). Pls expand on the epidemiological issues if available. CT scan in Fig 2 is 2 months after Tx. At this time the lady received full immunosuppression including basiliximab. Thus, did she become infected at that short period? Might be unlikely, but needs to be considered.

Discussion
L 114: I would even strengthen the statement and use the term "abortive or died-out" in order to indicate that a non-viable lesion becomes active during high-dose immunosuppression.
L 118: There is specific literature, mainly from Besancon and Berne, regarding the host-parasite-relationship. The mentioned citation is an attempt to clean the wording in the different settings, not to clarify the role of immunity.
L 121: There is a vast majority of papers with findings and opinions which were recently attempted to condense to a structured proposal Graeter et al. It is suggested to this paper as well as a recent paper in Pathogens by Grimm et al which is also related to the issue of this paper.
L 124: In the following paragraphs there are general statements without references, possibly as they are beyond the scope of the main message of this paper. 
L 133: How to get to a confirmed diagnosis of AE was laid out in Brunetti et al , Kern et al and Wen et al. Immunodiagnosis is an essential and important tool, but systematic immunoblotting is unrealistic, and it works only if serology is done at one site and by a most experienced laboratory as shown in citation #13. But it does not apply to general serological labs. Thus, this should be corrected and commented.
L 147: It is curious. AE may/is not of interest from the American point of view. In textbooks, the parasite E. granulosus is amply displayed, but AE is definitely neglected as AE does not occur in the Americas. May change! see recent paper in Clin Inf Dis. Therefore, reference to American guidelines may be misleading and wrong. Could be expanded on occasion of this paper?
L 149: Unfortunately, the last sentence about risk factors is not supported by data. Contaminated vegetables and fruits as a "dominant" risk factor belongs to the myths of echinoccosis, is displayed and re-iterated everywhere, but not supported by data (see above). To my opinion, final statement should cover the importance of dogs (less cats) as a proven risk factor. This includes a statement for preventive measures for dog owners, if authors really wish to expand. 

Kind of immunosuppression is not discussed for the specific case or of data from the literature. Perhaps, authors might wish to give more hints for the immunological background. Fact is that an "abortive = died-out = non-viable" lesion became reactivated despite assumed (literature Berne and Besancon) to be dead!!!

Abstract
L 33: Sentences are very general and not supported by data. Too much wording. Hypersensitivity test are rather unknown to general labs and impractical. I am suggesting to cut the abstract by half, at least.

Author Response

Dear Editor and Reviewers,

Many thanks for your constructive and supportive comments, for the time and attention you have dedicated to this manuscript. We have modified and completed our text according to your remarks. We believe it is now indeed more interesting than our first draft. It would be an honour to be published in your journal.
Best regards,

Damien Dupont (Corresponding Author)

Reviewer 1 Comment: This is an exciting case report, not only as it demonstrates the fact that immunosuppression leads to an accelerated growth of the parasite Echinococcus multilocularis, but more importantly, that parasitic lesions - identified as "aborted" or "non-viable"- also labelled as "died-out" in the literature - can be re-activated under certain circumstances.

Immunosusuppression and anti-GvHD treatment in this case led to a very fast growth of a parasitic lesion and demonstrates that "aborted lesions" can be still viable in humans. This finding may lead to reconsideration of immunological events after infection.

The paper is fluently written and expands on many aspects of the disease. However, some of the statements are more general and are not well referenced. Authors may decide to skip those parts and remain to the main results, or to expand with full references.

Authors' Response: We removed several sentences in the abstract and in the text that included general statements. Moreover, several references were added to expand introduction and discussion.

Reviewer 1 Comment: L 52: Explain the antibody basiliximab

Authors' Response: We explained that Basiliximab is a monoclonal antibody acting as a interleukin 2 receptor antagonist (L 66).

Reviewer 1 Comment: L 74: Histopathologically, AE was confirmed! This must be expressed. Liver biopsy is not recommended if the AE diagnosis is around! Please discuss later.

Authors' Response:According to the reviewer’s comments, we added additional data on histopathology (l. 90)“Microscopic examination of a liver biopsy fragment showed necrosis and presence of typical periodic acid-Schiff (PAS) positive thin lamellar layer, highly suggestive of AE” and “Pathological examination confirmed the diagnosis of AE, showing an unequivocal pseudotumoral lesion with ill-defined borders, numerous areas of necrosis and granuloma surrounding slender parasitic PAS-positive layers”.

We also highlighted that our case is a confirmed case, according to Brunetti’s consensus criteria. Finally, we highlighted the potential interest of liver biopsy for AE diagnosis, especially when serology is equivocal and negative. In our case, liver biopsy was performed before serological investigations as differential diagnosis for cholangiocarcinoma or metastasis.

A paragraph was added L 191 “This review also highlights the interest of histological examination and/or molecular techniques on liver biopsy, particularly in the same group of patients (negative serology but lesion shown with MRI/CT or US). Even if liver biopsy is not recommended as a first line approach, fine-needle biopsy is a minimally invasive and effective diagnostic to confirm hepatic AE [6,21]. Moreover, liver biopsy is not associated with major complications in AE patients. Histological examination of tissue, showing characteristic PAS-positive metacestode layer allows the diagnosis of AE, even if differential diagnosis with cystic echinococcosis is sometimes difficult. Thus , it should require implementation of immunohistochemistry assay using mAb Em2G11 staining.”

Reviewer 1 Comment: L 77: IHA is an unspecific test using E. granulosus antigen, often used as a screening test. It should not be stressed that the test was negative. This tells nothing. Specific tests are positive, and this is interesting as - often - under immunosuppression those test turn out to be negative! See Geyer et al. Discuss this fact.

Authors' Response: We agree with the reviewer's comment that IHA is only a screening test. However, a cut-off lowered to 1/80 gives a sensitivity of 94% for AE (Bart et al.. Diagnostic Microbiology and Infectious Disease 2007;59:93–95) and IHA positivity combined with a AE-compatible CT scan should lead to perform an immunoblot (Siles-Luacs et al. Advances Parasitolo. Indeed, in immunocompromised patients, all quantitative (Elisa) or semi-quantitative (IHA) techniques have significantly lowered sensitivity (Lachenmayer, Food and Waterborne Parasitology 2019 and Gottstein, Food and Waterborne Parasitology 2019). According to the reviewer’s comment, we deleted the sentence which previously stressed that IHA was negative in our patient.and we discussed later the reduced sensitivity of serological test in immunocompromised patient (l. 186): “In immunocompromised patients, first-line techniques (ELISA) have significantly lowered sensitivity and Em-immunoblotting appears as the most sensitive test to diagnose AE [19]. Similarly, a recent review on laboratory-based approach for AE diagnosis proposed implementation of immunoblotting as second line test in patient with compatible lesion shown by imaging techniques but with negative serology using first-line ELISA [20].

Reviewer 1 Comment: L 79: Diagnosis was confirmed by histopathology (see Brunetti et al, or Kern et al (not cited). Specific staining could have made the diagnosis more obvious (Barth et al.). Perhaps some additional words in the discussion? Could be an interesting issue.

Authors' Response: According to the reviewer’s comments, we added additional data on histopathology (l. 90): “Microscopic examination of a liver biopsy fragment showed necrosis and presence of typical periodic acid-Schiff (PAS) positive thin lamellar layer, highly suggestive of AE” and “Pathological examination led to confirmed diagnostic of AE, showing an unequivocal pseudotumoral lesions with numerous small vesicles and ill-defined borders, multiple areas of necrosis and granuloma surrounding slender parasitic PAS-positive layers”. We also discussed the histological differential diagnosis with cystic echinococcosis and the helpful contribution of immunohistochemistry using mAb Em2G11 (l. 195): “Histological examination of tissue, showing characteristic PAS-positive metacestode layer allows the diagnosis of AE, even if differential diagnosis with cystic echinococcosis is sometimes difficult. Thus , it should require implementation of immunohistochemistry assay using mAb Em2G11 staining [7,22].

Reviewer 1 Comment: L 84: Possibly, a PET/CT scan was not performed at the time of diagnosis? Should be mentioned why it was not considered (see Brunetti et al, Kern et al, Wen et al). Too costly, different clinical environment?

Authors' Response: A PET-CT Scan was indeed performed at the time of diagnosis that confirmed the existence of a 45 x 35 mm lesion with a central hypodense region and a strong peripheral hypermetabolic rim (Standardized uptake values max 5.3), persisting at 3 hours (SUV max 7.7). This was added in the text on line 88.

Reviewer 1 Comment: L 87: Drug levels are available. I am suggesting to add a table with the medication and level, as this gives much help to clinicians if confronted with similar cases. could also be discussed. Increased incidences of immunosuppression and AE is anticipated in the article of Chauchet et al.

Authors' Response: The range of values for albendazole sulfoxide were given in the text (L 106-108).

Reviewer 1 Comment: L 92: I am not convinced that such a weak band in IB would be suggestive of an abortive AE! Sera were checked in retrospect and with knowledge of the outcome! ELISA was negative. This should be the information of importance for readers and needs a change of the argumentation in Discussion.

Authors' Response: We agree with the reviewer's comment that we checked retrospectively the serum and that in many "all-round" patients, the immunoblot would not have been done. Em-Immunoblotting appears as the most sensitive test to diagnose inactive (“abortive”) AE-cases (as Elisa sensitivity was lower than 30% in such patients) in the study by Gottstein et al (Gottstein, Food and Waterborne Parasitology 2019). Moreover, serology using an extended panel of antigen and/or immunoblotting has been proposed as an alternative in patients with imaging suggestive of a parasitic lesion but with negative first-line serological tests (Siles-Lucas et al, Adv Parasitol. 2017).In our lab, in case of a patient with a suspected abortive AE lesion, we systematically perform an immunoblot even in case of first-line negative serological test, according to the French High Authority for Health recommendation (Haute Autorité de Santé, HAS) published in July 2017 (https://www.has-sante.fr/plugins/ModuleXitiKLEE/types/FileDocument/doXiti.jsp?id=c_2786936)

We discussed this specific point in the revised manuscript (L 186): “In immunocompromised patients, first-line techniques (ELISA) have significantly lowered sensitivity and Em-immunoblotting appears as the most sensitive test to diagnose AE [19]. Similarly, a recent review on laboratory-based approach for AE diagnosis proposed implementation of immunoblotting as second line test in patient with compatible lesion shown by imaging techniques but with negative serology using first-line ELISA.”

Reviewer 1 Comment: L 100: I miss the sentences regarding the activities of the veterinarian patient before and after tx. Did she has contact to her dogs while under immunosuppression? New- or re-infection issue needs to be discussed!! Had dogs/cats been treated with PZQ regularily, i.e. 6 weeks or around? Did she wash the vegetables from their garden? Statements in the paragraph are rather general and not adapted to the knowledge from literature (risk factors for AE, Kern et al). Pls expand on the epidemiological issues if available. CT scan in Fig 2 is 2 months after Tx. At this time the lady received full immunosuppression including basiliximab. Thus, did she become infected at that short period? Might be unlikely, but needs to be considered.

Authors' Response: The paragraph about risk factors was rewritten to include both pre and post transplantation periods and the risk factors found to be significant by Kern et al (l.122-129): “Direct interrogation at the time of diagnosis found exposure to several risk factors for AE prior to her transplantation and thereafter [10]: ownership of two dogs and four cats that were dewormed regularly but were free to roam outdoors and hunt unattended including in her unfenced garden visited by foxes; living in a house close to fields, vocational activities in the forest and growing leaf and root vegetable. She worked as a veterinarian for pets prior to transplantation but did not resume work before seven months post transplantation. She reported washing produces from her garden and good hand hygiene after handling and playing with dogs and cats.

Reviewer 1 Comment: L 114: I would even strengthen the statement and use the term "abortive or died-out" in order to indicate that a non-viable lesion becomes active during high-dose immunosuppression.

Authors' Response: We agree with the reviewer’s comment: the term abortive mean (presumptively) dead parasite. However, as it was only based on imaging data, it cannot be excluded that several parasitic stem cell remained alive, playing the role of trojan horse. We highlighted this point in the revised manuscript. According to recent consensus on Echinococcus terminology (Vuitton et al, 2020), we used the term aborted lesion instead of abortive or died-out lesion in the revised mansucript.

Reviewer 1 Comment: L 118: There is specific literature, mainly from Besancon and Berne, regarding the host-parasite-relationship. The mentioned citation is an attempt to clean the wording in the different settings, not to clarify the role of immunity.

Authors' Response: Additonal information has been added from line 149: “The natural history of AE is the consequence of a balance between the growth of the metacestode and the host's response, which depends on the genetic background of the host as well as on acquired disturbances of Th1-related immune response. Th1-oriented immune response induces protective immunity, leading eventually to self-limited lesion. The laminated layer plays a major role in the induction of tolerance, driven by Th2 response and anti-inflammatory cytokines, especially IL-10 and TGF-β [2,15]. Our patient received several immunosuppressive treatment that significantly impact on Th1/Th2 balance and thus host-parasite interaction, mainly tacrolimus, calcineurin inhibitor known to inhibit the T-cell response and decrease Th1/Th2 ratio and basiliximab, antagonist of IL2-receptor. Small lesion with central calcification, as observed in our patient, fit with the description of CT type IV lesion using EMUC CT classification [16]. Some authors hypothesize that this stage corresponds to the initial phase of the infection, with a strong immune response around the lesion to contain the infection, leading to the persistence of a self-limited dormant infection. Subsequently, this stage can evolve either towards an active infection (EMUC stage I to III) or involute, with necrosis and/or increased calcification [16,17].

Reviewer 1 Comment: L 121: There is a vast majority of papers with findings and opinions which were recently attempted to condense to a structured proposal Graeter et al. It is suggested to this paper as well as a recent paper in Pathogens by Grimm et al which is also related to the issue of this paper.

Authors' Response: According to these two comments, this paragraph was rewritten in the revised manuscript (l.158 and so on): “Some authors hypothesize that this stage corresponds to the initial phase of the infection, with a strong immune response around the lesion to contain the infection, leading to the persistence of a self-limited dormant infection. Subsequently, this stage can evolve either towards an active infection (EMUC stage I to III) or involute, with necrosis and/or increased calcification [16,17]., They can have different aspects but usually include one or several small size calcifications.. Active AE imaging patterns vary from small cystoid lesion to primarily circumscribed tumor-like or diffuse infiltrating lesion with various degree of calcifications using CT-scan. The pathognomonic honeycomb-like AE microcysts were shown using T2 weighted MRI [10,16].

Reviewer 1 Comment: L 124: In the following paragraphs there are general statements without references, possibly as they are beyond the scope of the main message of this paper. 

Authors' Response: We shortened the text.

Reviewer 1 Comment: L 133: How to get to a confirmed diagnosis of AE was laid out in Brunetti et al , Kern et al and Wen et al. Immunodiagnosis is an essential and important tool, but systematic immunoblotting is unrealistic, and it works only if serology is done at one site and by a most experienced laboratory as shown in citation #13. But it does not apply to general serological labs. Thus, this should be corrected and commented.

Authors' Response: Immunoblotting could be helpful in selected cases when first-line tests are negative (immunocompromised patients with lesion, even if atypical, or patient with small/calcified lesion). This was added in in the revised manuscript.

Reviewer 1 Comment: L 147: It is curious. AE may/is not of interest from the American point of view. In textbooks, the parasite E. granulosus is amply displayed, but AE is definitely neglected as AE does not occur in the Americas. May change! see recent paper in Clin Inf Dis. Therefore, reference to American guidelines may be misleading and wrong. Could be expanded on occasion of this paper?

Authors' Response: The reviewer is fully right. We removed the specific reference to American guidelines and called for national initiatives and guidelines wherever epidemiologically relevant.

Reviewer 1 Comment: L 149: Unfortunately, the last sentence about risk factors is not supported by data. Contaminated vegetables and fruits as a "dominant" risk factor belongs to the myths of echinoccosis, is displayed and re-iterated everywhere, but not supported by data (see above). To my opinion, final statement should cover the importance of dogs (less cats) as a proven risk factor. This includes a statement for preventive measures for dog owners, if authors really wish to expand. 

Authors' Response: We modified the text and mentioned that prevention should target dog owners (l. 209): “Information should more specifically target dog owners [10] and could be relayed by veterinarians and pet groomers, with the support of educational flyers or posters addressing the need for regular deworming, for preventing roaming and for good hand hygiene after any contact.

Reviewer 1 Comment: Kind of immunosuppression is not discussed for the specific case or of data from the literature. Perhaps, authors might wish to give more hints for the immunological background. Fact is that an "abortive = died-out = non-viable" lesion became reactivated despite assumed (literature Berne and Besancon) to be dead!

Authors' Response: “The natural history of AE is the consequence of a balance between the growth of the metacestode and the host's response, which depends on the genetic background of the host as well as on acquired disturbances of Th1-related immune response. Th1-oriented immune response induces protective immunity, leading eventually to self-limited lesion. The laminated layer plays a major role in the induction of tolerance, driven by Th2 response and anti-inflammatory cytokines, especially IL-10 and TGF-β [2,15]. Our patient received several immunosuppressive treatment that significantly impact on Th1/Th2 balance and thus host-parasite interaction, mainly tacrolimus, calcineurin inhibitor known to inhibit the T-cell response and decrease Th1/Th2 ratio and basiliximab, antagonist of IL2-receptor. » was added from line 149.

Reviewer 1 Comment: L 33: Sentences are very general and not supported by data. Too much wording. Hypersensitivity test are rather unknown to general labs and impractical. I am suggesting to cut the abstract by half, at least.

Authors' Response: We cut the abstract by half, cutting the last sentences.

Reviewer 2 Report

First I absolutely enjoyed to read and review this article.

I would suggest to re-think about the title (primarily the term unexpectedly) because fast growing of an AE liver lesion is not that unexpected under immunosuppression after solid organ transplanatation.

Abstract:

Line 30: "the preexistence of abortive self-limited lesions of AE" the sentence is correct so far BUT the abortive lesion must be explained in detail (in text). What is the exact definition of abortive AE-lesion and how to proof it?

Line 39: “educate patients at risk on how to avoid”…seems a good idea but I would say from my experience “easy to recommend, impossible to follow”: I mean YES education on hygiene behavior belongs to transplant aftercare. What exactly would the authors suggest to tell patients?   BUT more or most important is to educate involved physicians in aftercare and awareness about AE as one possible reasons for liver lesions in immunosuppressed patients.

Abstract Last Sentence: Again the authors should please provide exact definition on “abortive AE lesions” and how to deal with them in terms of diagnostic and treatment.

The definition of slow and fast growing in this context should be given.

Case:

Line 70: “fortunately, just before initiation of chemotherapy” REALLY what chemotherapy without histology???

Line 78: Immunoblotting confirmed the diagnosis of AE: not correct because immunoblotting is NOT a confirmatory test in terms of WHO-IWGE definition, it is just a serological confirmation test BUT confirmation of AE diagnosis according case definition Brunetti t al 2010 Acta tropica is PCR or histology of a clinical sample.

Line 91: Interesting…is 7-band in IB really compatible with abortive, self-limited AE. Could the authors please provide again conditions for “self-limiting” AE.

Discussion:

Line 109: Lit [6] is a review Wen H: please provide primary literature source

Line 148-150: Again as told in the abstract, prevention is important and patients education, but avoid exposition seems almost impossible BUT awareness of physicians in detection lesions and perform diagnostics and treatment is of great importance. Exposition to E.m. might have been long before initiation the immunosuppression!

Line 146, 147: "probably" is written twice in sentence

Altogether an interesting case and a good article! Go on!

Author Response

Reviewer 2 Comment: First I absolutely enjoyed to read and review this article.

I would suggest to re-think about the title (primarily the term unexpectedly) because fast growing of an AE liver lesion is not that unexpected under immunosuppression after solid organ transplantation.

Authors' Response: Thanks. We fully agree with the reviewer and removed the word “unexpectedly” from the title.

Reviewer 2 Comment: Line 30: "the preexistence of abortive self-limited lesions of AE" the sentence is correct so far BUT the abortive lesion must be explained in detail (in text). What is the exact definition of abortive AE-lesion and how to proof it?

Authors' Response: We added a specific paragraph on natural history of AE. Moreover, definition of aborted lesion was given as requested (see above).

Reviewer 2 Comment: Line 39: “educate patients at risk on how to avoid”…seems a good idea but I would say from my experience “easy to recommend, impossible to follow”: I mean YES education on hygiene behavior belongs to transplant aftercare. What exactly would the authors suggest to tell patients?   BUT more or most important is to educate involved physicians in aftercare and awareness about AE as one possible reasons for liver lesions in immunosuppressed patients.

Authors' Response: We agree with the reviewer. The abstract was shortened, but information was added in the text line 209 regarding the need to target the message to dog owners and to involve physicians in prevention: ““Information should more specifically target dog owners [10] and could be relayed by veterinarians and pet groomers, with the support of educational flyers or posters addressing the need for regular deworming, for preventing roaming and for good hand hygiene after any contact. 

Reviewer 2 Comment: Abstract Last Sentence: Again the authors should please provide exact definition on “abortive AE lesions” and how to deal with them in terms of diagnostic and treatment.

The definition of slow and fast growing in this context should be given.

Authors' Response: Modification were made accordingly (see above). Definition of slow and fast was added line 139.

Reviewer 2 Comment: Line 70: “fortunately, just before initiation of chemotherapy” REALLY what chemotherapy without histology?

Authors' Response: We removed this information on the referral to the oncology department that had already been scheduled.

Reviewer 2 Comment: Line 78: Immunoblotting confirmed the diagnosis of AE: not correct because immunoblotting is NOT a confirmatory test in terms of WHO-IWGE definition, it is just a serological confirmation test BUT confirmation of AE diagnosis according case definition Brunetti t al 2010 Acta tropica is PCR or histology of a clinical sample.

Authors' Response: We corrected this mistake. Confirmed AE diagnosis was based on histology. Serology was concordant with this diagnosis, with IB pattern often shown in AE case. The serology allows at best to describe a probable AE case, according to Brunetti's criteria.

Reviewer 2 Comment: Line 91: Interesting…is 7-band in IB really compatible with abortive, self-limited AE. Could the authors please provide again conditions for “self-limiting” AE.

Authors' Response: As previously requested by reviewer 1, we added specific paragraph of natural history of AE.

Reviewer 2 Comment: Line 109: Lit [6] is a review Wen H: please provide primary literature source

Authors' Response: Modification was made accordingly by adding Amman et al 1996.

Reviewer 2 Comment: Line 148-150: Again as told in the abstract, prevention is important and patients education, but avoid exposition seems almost impossible BUT awareness of physicians in detection lesions and perform diagnostics and treatment is of great importance. Exposition to E.m. might have been long before initiation the immunosuppression!

Authors' Response: The need to raise awareness among physicians was stressed line 175-178: “The worldwide extension of areas of active transmission and the possible resurgence in historic foci, should lead to increased awareness, and to listing AE among the etiologies for liver lesions in national education programs and guidelines regarding transplantation and other contexts involving immunosuppression, wherever epidemiologically relevant.

Reviewer 2 Comment: Line 146, 147: "probably" is written twice in sentence.

Authors' Response: Modification has been made accordingly.

Round 2

Reviewer 1 Report

Authors have responded to all major points and modified the manuscript accordingly. Well done.

Just an additional point. Line 160. Authors have described the pos. PET/CT in the case report. Here, in Discussion, this would be a proof of the statement of their own case. Add a sentence.

Another interesting point is the time sequence of immunosuppression and appearence of the lesion. In the two months after lung tx the patient received full immunosuppression which was possibly tapered later. Here the lesion might have been exploded. At the time of diagnosis (11 months post-tx), the AE lesion presents as shown for most of the (ii) cases (see natural history paragraph). Thus, PET/CT was indicative, and led to images as for most "regular" AE cases, showing strong cellular reaction in the surrounding of the lesion. The further dissection of the immune markers in the follow-up after tx might be beyond this paper, but would be a most interesting aspect for further studies.

Author Response

Reviewer 1 Comment: Authors have responded to all major points and modified the manuscript accordingly. Well done.

Just an additional point. Line 160. Authors have described the pos. PET/CT in the case report. Here, in Discussion, this would be a proof of the statement of their own case. Add a sentence.

Authors' Response: Sentence has been added to the discussion (l. 160) « The FDG-PET/CT often shows increased uptake of FDG surrounding AE lesions higher than in other areas (as in our patient), and has thus become the preferred reference tool for assessing their metabolic activity (Kern 2017, Wen 2019).

Reviewer 1 Comment: Another interesting point is the time sequence of immunosuppression and appearence of the lesion. In the two months after lung tx the patient received full immunosuppression which was possibly tapered later. Here the lesion might have been exploded. At the time of diagnosis (11 months post-tx), the AE lesion presents as shown for most of the (ii) cases (see natural history paragraph). Thus, PET/CT was indicative, and led to images as for most "regular" AE cases, showing strong cellular reaction in the surrounding of the lesion. The further dissection of the immune markers in the follow-up after tx might be beyond this paper, but would be a most interesting aspect for further studies.

Authors' Response: Although host-parasite interactions during natural infection are now better understood, the precise impact of immunosuppressive drugs on these interactions and on the pathophysiology of AE in immunocompromised patients are rather poorly known. A better knowledge of these complex interactions would allow a better understanding of the rapid evolution of lesions and their sometimes baffling clinical-radiological presentations. We agree with the reviewer that this is beyond our manuscript, as this would require either studies on animal models and/or prospective follow-up of SOT patients with a very low incidence of infection.